# The Influence of Physiological Blood Clot on Osteoblastic Cell Response to a Chitosan-Based 3D Scaffold—A Pilot Investigation

**DOI:** 10.3390/biomimetics9120782

**Published:** 2024-12-21

**Authors:** Natacha Malu Miranda da Costa, Hilary Ignes Palma Caetano, Larissa Miranda Aguiar, Ludovica Parisi, Benedetta Ghezzi, Lisa Elviri, Leonardo Raphael Zuardi, Paulo Tambasco de Oliveira, Daniela Bazan Palioto

**Affiliations:** 1Department of Oral and Maxillofacial Surgery and Periodontology, School of Dentistry of Ribeirão Preto, University of São Paulo, Avenida Do Café-Subsetor Oeste-11 (N-11), Ribeirão Preto 14040-904, SP, Brazil; natachamalu@gmail.com (N.M.M.d.C.); hilary.caetano@usp.br (H.I.P.C.); larissa_m.aguiar@hotmail.com (L.M.A.); 2Laboratory for Oral Molecular Biology, Department of Orthodontics and Dentofacial Orthopedics, University of Bern, Freiburgstrasse 3, 3010 Bern, Switzerland; ludovica.parisi@unibe.ch; 3Centro Universitario di Odontoiatria, Dipartimento di Medicina e Chirurgia, University of Parma, Via Gramsci 14, 43126 Parma, Italy; benedetta.ghezzi@unipr.it; 4Istituto dei Materiali per l’Elettronica ed il Magnetismo, Consiglio Nazionale Delle Ricerche, Parco Area Delle Scienze 37/A, 43124 Parma, Italy; lisa.elviri@unipr.it; 5Department of Basic and Oral Biology, School of Dentistry of Ribeirão Preto, University of São Paulo, Avenida Do Café-Subsetor Oeste-11 (N-11), Ribeirão Preto 14040-904, SP, Brazil; leonardo.zuardi@usp.br (L.R.Z.); tambasco@usp.br (P.T.d.O.)

**Keywords:** blood clot, ex vivo assay, three dimensional, scaffold, hydrogel, osteogenesis

## Abstract

Background: The use of ex vivo assays associated with biomaterials may allow the short-term visualization of a specific cell type response inserted in a local microenvironment. Blood is the first component to come into contact with biomaterials, providing blood clot formation, being substantial in new tissue formation. Thus, this research investigated the physiological blood clot (PhC) patterns formed in 3D scaffolds (SCAs), based on chitosan and 20% beta-tricalcium phosphate and its effect on osteogenesis. Initially, SCA were inserted for 16 h in rats calvaria defects, and, after that, osteoblasts cells (OSB; UMR-106 lineage) were seeded on the substrate formed. The groups tested were SCA + OSB and SCA + PhC + OSB. Cell viability was checked by MTT and mineralized matrix formation in OSB using alizarin red (ARS). The alkaline phosphatase (ALP) and bone sialoprotein (BSP) expression in OSB was investigated by indirect immunofluorescence (IF). The OSB and PhC morphology was verified by scanning electron microscopy (SEM). Results: The SCA + PhC + OSB group showed greater cell viability (*p* = 0.0169). After 10 days, there was more mineralized matrix deposition (*p* = 0.0365) and high ALP immunostaining (*p* = 0.0021) in the SCA + OSB group. In contrast, BSP was more expressed in OSB seeded on SCA with PhC (*p* = 0.0033). Conclusions: These findings show the feasibility of using PhC in ex vivo assays. Additionally, its inclusion in the experiments resulted in a change in OSB behavior when compared to in vitro assays. This “closer to nature” environment can completely change the scenario of a study.

## 1. Introduction

In vitro experiments are routine in research associated with the use of biomaterials when studying responses to basic cellular mechanisms in a quantitative way [1]—they are usually applied in pre-clinical trials due to their practicality, accessibility, and ease of acquiring results [2]. One of the reasons they are frequently used is because in vivo studies are less subject to ethical issues, which can sometimes be challenging and conflicting [2]. Unfortunately, most in vitro models do not reflect the interaction of host responses, limiting the reliability of the results.

Ex vivo assays were previously proposed as an interesting method to better understand biomaterial short-term response, profile, and performance. Most of all, this model aims to acquire the knowledge of events that happen, naturally, after the insertion of a biomaterial in a given tissue microenvironment [3,4]. The interactions between different cellular components trigger diverse in situ autocrine and paracrine signaling pathways [3,5].

The initial events associated with healing substantially depend on the physiological blood clot (PhC) components of the host [6]—in such cases, blood leads to clot formation, serving as a support and substrate for new tissue formation [7]. The biological phenomenon of blood clotting comprises a series of chemotactic and biomolecular events orchestrated by different cell types [6,8]. The goal of blood coagulation is to produce hemostasis through a fibrillar mesh and platelet structure [7], serving as a three-dimensional (3D) biological meshwork for cell recruitment related to local repair [9,10].

The use of 3D scaffolds (SCA) was demonstrated to overcome the limitations of two-dimensional in vitro assays [11]. Three-dimensional SCA provide a homogeneous cell distribution [12] and, consequently, high cell conductivity [13]. The 3D SCA organization ensures the propagation of nutrients and oxygen, mainly due to the absence of blood vessels in the grafts [14,15]. In addition, cell–cell or cell–extracellular matrix (ECM) contact occurs in a more harmonious way, as in the natural environment [15]. However, a disadvantage of using 3D SCA is the presence of only one constituent in its structure, generally similar to the ECM, interacting with a specific cell [2]. This completely distorts the physiological response that takes place in an immunoinflammatory and reparative microenvironment. Several materials were developed for the preparation of these 3D SCA [16]. Among them are biodegradable polymers, such as hydrogel [17].

Hydrogels are highly hydrophilic compounds with excellent biological properties. They can be composed of natural polymers, such as those derived from ECM components (e.g., collagen, gelatin, fibrin, hyaluronic acid) or seaweed sources (e.g., alginate, agarose, chitosan). Alternatively, they can be made from chemical or synthetic polymers, including those derived from ethylene oxide, vinyl alcohol, acrylic acid, propylene fumarate-co-ethylene glycol, and polypeptides [17].

In this context, natural biodegradable polymers are the preferred choice over non-biodegradable ones due to their reduced toxicity, rapid biodegradation, osteogenic potential, and high biocompatibility. Conversely, non-biodegradable polymers, while associated with some toxicity and inflammatory responses, offer a slower degradation rate and superior physical–mechanical properties [18].

Among natural hydrogels, chitosan was extensively studied for its potential in developing 3D scaffolds for bone regeneration [19]. Due to its mechanical properties and composition resembling the ECM, chitosan proved to be an excellent biomaterial for creating membranes used in guided tissue regeneration in periodontology [20]. Moreover, it exhibits effective antimicrobial activity against various oral pathogens [21].

Thus, the purpose of this study was to compare bone cell features development on chitosan-based 3D SCA in two different models of analysis: in vitro and ex vivo. This PhC was formed in a critical defect created in rat calvaria. Additionally, the effect of this PhC on osteogenesis was also verified in a 3D microenvironment, using UMR-106 osteoblastic cell lineage.

## 2. Materials And Methods

### 2.1. Ethics Aspects

This research project was submitted to Committee of Ethics in Animal Research of the School of Dentistry of Ribeirao Preto, University of Sao Paulo, which approved all the animal procedures performed (process number 2018.1.247.58.7). This article was further written in accordance with the ARRIVE guidelines [22].

A total of 42 male Wistar Hannover rats, 12 weeks old, were used in this study. The rats were obtained from the Central Animal Facility of the Ribeirão Preto Campus at the University of São Paulo (USP). These animals were chosen due to their anatomical, physiological, and genetic similarities to humans [23]. The sample size was determined using GraphPad Prism software (5.0 version, GraphPad Software Inc., La Jolla, CA, USA). To ensure 80% statistical power, the ideal sample size was calculated based on the mean differences and standard deviations, with the α value set at 0.05 [24]. The rats were randomly housed in plastic vivarium cages, with four animals per cage. During the housing period, they were provided with a balanced diet and had access to water ad libitum. They were kept at a controlled temperature of 25 °C under a 12-h light/dark cycle.

### 2.2. 3D Chitosan/β-TCP Scaffolds

The fabrication of chitosan-based 3D SCA and 20% beta-tricalcium phosphate (β-TCP), was performed in a built in-house LTM system using the structure of a commercial Fuse Deposition (FDM) manufacturing 3D printer [25], producing an SCA with a porosity of 200 µm. In this work, Chitosan (ChitoClear™ TM4030, 50 kDa, Primex Ehf, Siglufjörður, Iceland) with a deacetylation degree of 75% was used. Acetic acid 99.8% was obtained from Sigma-Aldrich (St. Louis, MO, USA). All solutions were prepared using ultrapure water obtained with a Purelab Flex 1 system (ELGA Veolia, Lane End, UK). Chitosan was dissolved in ultrapure water at a 6% *w*/*v* concentration with the help of 2% *v*/*v* of acetic acid under stirring. After the complete dissolution of chitosan (Primex Ehf), the calcium phosphate (≥95%, Sigma-Aldrich) was added and mixed until complete homogenization before printing.

The ink solution was loaded into a 10 mL syringe that was accommodated on the 3D printer [25]. Briefly, the instrument exploited a three-Cartesian-axis system that allows the movement on the horizontal plane of the printing plate and the vertical translation of the nozzle. The chitosan-based solution is printed by a pump-acting extrusion process throughout a syringe mounting a 26 G needle. A layer-by-layer deposition mode is carried out on the printing plate at a velocity of 3 mm/s and instantaneously frozen (−10 °C) to keep the SCA structure. SCA (1.6 cm × 1.6 cm in size) were composed by the alternation in the orthogonal layers of parallel filaments, being 200 µm distant from each other.

After freezing the 3D-printed SCA, the structure was fixed by exposition to ammonia (2.0 M in ethanol, Sigma-Aldrich) vapor in a saturated chamber. This part of the methodology was carried out in partnership with the Department of Biomedical, Biotechnological and Translational Sciences at the University of Parma (Parma, IT), as previous described [26].

### 2.3. Osteoblastic Line

Pre-osteoblastic cell culture from UMR-106 cell line (Osteoblast – OSB, (American Type Culture Collection, ATCC CRL-1661, Manassas, VA, USA), derived from rats, preserved in 2 mL cryogenic tubes (Corning Inc., Corning, NY, USA) in liquid nitrogen drums, were used in cell proliferation, ECM synthesis, and ECM mineralization assays. These cells were manipulated in type II laminar flow (São Bernardo do Campo, SP, Brazil) and cultured in 75 cm^3^ cell culture flasks (Corning Inc.) with 10 mL of Dulbecco’s Modified Eagle Medium (D-MEM, Invitrogen, Thermo Fisher Scientific, Carlsbad, CA, USA), supplemented with 10% fetal bovine serum (FBS, Invitrogen) and 2.2 mL of penicillin/streptomycin solution at 100 µg/mL and 100 IU/mL, respectively (Gibco, Thermo Fisher Scientific, Waltham, MA, USA). During the entire culture time, the cells were kept at 37 °C in a humidified atmosphere, containing 5% CO_2_ and 95% atmospheric air. Media were changed every 2–3 days. Cells growth was monitored daily under an Axiovert 40C inverted phase contrast microscope (Carl Zeiss, Oberkochen, BW, Germany). All experiments were performed in biological triplicate, with two replications.

### 2.4. “Physiological” Blood Clot Formation

For ex vivo blood clot formation, male Hannover rats, aged 12 weeks, acquired at the Central Biotherm of USP Campus of Ribeirão Preto were used. The surgical procedure for research developing was performed using a modified experimental protocol as previously described [24,26,27]. Initially, the animals were anesthetized with Xylazine (10 mg/Kg, Syntec, Barueri, SP, Brazil) and Ketamine (80 mg/Kg, Syntec) intramuscularly (IM). After reaching a deep anesthetic plane, the animals were placed in the prone position on a surgical table, and trichotomy was performed in the dermis adjacent to the calvaria, followed by antisepsis with Polyvinyl Pyrrolidone Iodine (10%, São José do Rio Preto, SP, Brazil), and a semilunar incision was performed with a 15C scalpel (Solidor, São Paulo, SP, Brazil). The dermis was detached along with the periosteum using a Molt 2–4 detacher (Hu-Friedy, Chicago, IL, USA). With the calvaria exposed, an electric motor BLM 500 VK (Driller, Carapicuíba, SP, Brazil) was used to create the bone defect, with a 16:1 reduction contra-angle at a speed of 800 rotations per minute (rpm), coupled with a drill. 5 mm diameter trephine (Quinelato, Rio Claro, SP, Brazil) and abundant irrigation with sterile saline solution. During surgery, the bone segment was gently removed with a Molt 2–4 detacher (Hu-Friedy), in order to keep the dura mater intact, since this membrane is rich in progenitor cells [28]. After creating the critical defect in the calvaria, the 5 mm diameter and 4 mm thick SCA were positioned in the bone defect. The incision was sutured with Nylon 4.0 thread (Ethicon, New Jersey, NY, USA). When surgery was completed, the animal received the analgesic tramadol hydrochloride Cronidor^TM^ (2 mg/kg, Agener União, Taboão da Serra, SP, Brazil), subcutaneously, being monitored until the return of its corneal reflexes and kept in individual cages for the formation of the physiological blood clot (PhC) in the SCA inserted into the calvaria. After a period of 16 h, the animals were euthanized by an anesthetic overdose of Lidocaine (0.7 mg/Kg, 10 mg/mL, Syntec) associated with sodium thiopentate 2.5% (150 mg/kg, Syntec) to remove the biomaterial together with the PhC and subsequent ex vivo assay. The PhC removed from the calvaria was kept in a phosphate buffer (PB) solution (Gibco) and directed to cell culture assays.

### 2.5. Ex Vivo Assay

#### 2.5.1. Flow Citometry

The cell population characterization present in PhC formed on SCA was performed by flow cytometry at the Laboratory of High Resolution Images and Cellular Studies (LIAREC) of School of Pharmaceutical Sciences of Ribeirão Preto (FCFRP-USP). Initially, the PhC was formed as previously described in the surgical protocol. Then, the PhC was gently washed with PB (Gibco) in a 24-well plate (Corning Inc.) at room temperature ( RT). To remove the PhC cells adhered on SCA, the substrate (SCA + PhC) was treated with 5% trypsin (Gibco) and ethylenediaminetetraacetic acid (1 mM EDTA solution; Gibco) for 5 min at RT. The blockade of trypsin enzymatic activity was performed by means of D-MEM (Gibco) supplemented with 10% FBS (Gibco), at RT. The removed cells were added to a 15 mL Falcon tube (Corning Inc.) and centrifuged at 2000 rpm for 5 min. The pellet formed in the tube was treated with eBioscience^TM^ 1X RBC Lysis Buffer (Thermo Fisher Scientific) for 10 min at RT. The remaining cells (white blood cells) were separated into Falcon tubes (Corning Inc.) for flow cytometry (BD-Biosciences Inc., New Jersey, NY, USA) at a density of 1 × 10^5^ cells/tube.

For the characterization of the PhC cells, the primary monoclonal antibodies anti-CD45 (1 µg/1 × 10^6^ cells, CD45/PE, BD-Biosciences), anti-CD90 (1 µg/1 × 10^6^ cells, CD90/FITC, BD-Biosciences), anti-CD34 (1 µg/1 × 10^6^ cells, CD34/PE- Santa Cruz Biotechnology, Dallas, TX, USA), anti-CD44 (1 µg/1 × 10^6^ cells, CD44/FITC, BD-Biosciences), anti-CD42 (1 µg/1 × 10^6^ cells, CD42/FITC, BD-Biosciences), and anti-CD61 (1 µg/1 × 10^6^ cells, CD61/FITC, BD-Biosciences), diluted in PB (Gibco), were incubated for 30 min, RT. The selected antigens are specific surface markers for T and B lymphocytes, mesenchymal stem cells (MSCs), hematopoietic progenitor cells, leukocytes, platelets, and the Von Willebrand factor of endothelial cells, respectively. Then, the samples were fixed with 4% Paraformaldehyde (Merck Millipore, Burlington, MA, USA), diluted in PB (1:1, Gibco), and evaluated by the BD FACsCanto™ II flow cytometer (BD Biosciences). The analyses were performed using the BD FACSDiva™ 5.0 software (v9.0, 64-bit, BD Biosciences, New Jersey, NY, USA), and the results were given in a percentage of labeled cells for each antibody.

#### 2.5.2. Cell Culture on Substrate (SCA + PhC)

After reaching around 80% confluence in the culture bottles, the OSB were removed from the culture bottles by enzymatic dissociation by means of treatment with 1 mM EDTA (Gibco) and 0.25% trypsin (Gibco), and subsequently counted under an inverted microscope (Carl Zeiss), with the aid of a hemocytometer (Thermo Fisher Scientific). Cells were grown on SCA contained in 96-well plates (Corning Inc.), at a density of 5 × 10^4^ cells/well, and cultured in D-MEM (Gibco) culture medium containing 10% FBS (Gibco), 5 µg/mL ascorbic acid (Gibco) and 7 mM of beta-glycerophosphate (Gibco), being kept in a humid atmosphere with 5% of CO_2_ and 95% of atmospheric air.

#### 2.5.3. Cell Viability

The MTT [3-(4,5-dimethylthiazol-2-yl)-2,5-diphenyltetrazolium bromide] (Sigma-Aldrich), a salt that is reduced by mitochondrial proteinases, active only in viable cells [29]. This assay was performed for 72 h, as it is an ideal time to check cell viability [30]. Aliquots of MTT at 5 mg/mL in phosphate buffered saline (PBS, Invitrogen) were prepared and added to the culture medium, remaining at a concentration of 10% for 4 h at 37 °C, being maintained in a humid atmosphere containing 5% CO_2_ and 95% of atmospheric air. After this period, the cultures were washed with 1 mL of warm PBS (Invitrogen). Then, 1 mL of acid isopropanol solution, prepared with 100 mL of isopropanol (99%, Sigma-Aldrich) and 134 µL of HCl (37%, Sigma-Aldrich), was added to each well under stirring for 5 min, for the complete solubilization of the formed precipitate. Aliquots of 200 µL were taken from the wells and transferred to a 96-well plate (Corning Inc.) for colorimetric measurement in a spectrophotometer (570 nm; µQuanti, BioTek Instruments Inc., Charlotte, VT, USA).

#### 2.5.4. Scanning Electron Microscopy

The morphology of the OSB and PhC was accessed by scanning electron microscopy (SEM) at Multiuser Electron Microscopy Laboratory (LMME) of Ribeirão Preto Medical School (FMRP-USP). Initially, the OSB seeded on the SCA with and without PhC were maintained for 72 h in the culture medium, as previously mentioned. The substrate formed on the SCA was fixed in a 2.5% glutaraldehyde solution (50%, Merk Millipore) in 0.2 M cacodylate buffer (50 mM— Isopropanol 15%, Merk Millipore). The experimental groups investigated were three: SCA + OSB, SCA + PhC + OSB and SCA + PhC. The samples were dehydrated in increasing concentrations of ethanol (30%, 50%, 70%, 90%, 95%, and absolute, Merck Millipore) and metallized in a gold bath for 120 s by Bal-Tec SCD 050 (Baltec, Pfäffikon, CH). The acquisition of images in the respective groups was performed using the SEM equipment Jeol JSM -6610 LV (Jeol, Akishima, Tokyo, JPN), with a resolution of 5.0 nm at 25 kV, WD10 mm, and magnification from 15× to 200,000×.

#### 2.5.5. Detection of Calcium Accumulations (Mineralized Matrix Formation)

OSB cultured on SCA and SCA + PhC for 7 and 10 days were washed with Hanks’ solution, fixed in 70% ethanol (Invitrogen) for 60 min (4 °C), and washed with PBS (Gibco) and double-distilled water. Then, they were stained with 2% Alizarin Red Stain (ARS) (Sigma-Aldrich), pH 4.2 for 15 min (RT), washed with PBS (Gibco) and double-distilled water, and allowed to dry at RT. Morphologically, the proportions of areas marked with ARS were evaluated by the Leica QWin program (3.0 version, Leica Microsystems, Cambridge, UK), by means of color detection by intensity, from images obtained with a Leica MZ6 stereo-microscope (Leica Microsystems, Wetzlar, Germany), objective 8X, using a Leica DC300 F (Leica) camera of 1.3 MPixel resolution. Biochemically, calcium accumula [31]. In each well of the 96-well plate (Corning Inc.) containing the ARS-stained samples, 280 µL of 10% acetic acid ( ≥99.7%, Sigma-Aldrich) were added under gentle agitation for 30 min. The cell layer removed with a scraper and the SCA were transferred together with the acid solution into 1.5 mL Eppendorf tubes (Corning Inc.) and vortexed for 30 s. Then, the samples were heated at 85 °C for 10 min, cooled on ice for 5 min, and centrifuged at 13,000× *g* for 20 min. From each experimental group, 100 µL of supernatant was transferred to a 96-well plate (Corning Inc.) and 40 µL of 10% ammonium hydroxide was added to each well. Samples were read in a µQuanti spectrophotometer (BioTek Instruments Inc.) at 405 nm. The standard curve was performed with successive dissolutions of 0.5 to 3 mM ARS in ammonium acetate (10% acetic acid and 5 M ammonium hydroxide).

#### 2.5.6. Cell Morphology and Three-Dimensional Analysis of Substrate by Confocal Microscopy

This methodology was used to verify the labeling of alkaline phosphatase (ALP) and bone sialoprotein (BSP) OSB on SCA and SCA + PhC, the protocol being adapted from De Oliveira and Nanci (2004) [32]. After 10 days, the OSB on the SCA with and without PhC were washed with PB (Gibco, Invitrogen) and fixed with 4% paraformaldehyde at 0.1 M PB, pH 7.2, for 10 min (RT). Cell membrane cells were permeabilized with 0.5% Triton x-100 solution in PB (Gibco) for 10 min, followed by blocking with 5% goat serum in PB for 30 min. For the detection of ALP and BSP, primary antibodies to anti-ALP (rabbit, 1:100, Bioss, Massachusetts, USA) and anti-BSP (mouse, 1:200, Developmental Studies Hybridoma Bank, Iowa City, IA, USA) were incubated for 60 min, RT. For the detection of primary antibodies, Alexa Fluor^TM^ 594 (1:200, Molecular Probes, Eugene, OR, USA) and Alexa Fluor^TM^ 488 (1:200, Molecular Probes) anti-mouse (1:200, Molecular Probes), diluted in PB (Gibco), were used for 50 min (RT). The cell nucleus was labeled with DAPI (Molecular Probes) at 300 nM for 5 min. The SCAs were mounted between two 24 × 32 mm glass coverslips (Knittel Glass, Braunschweig, Germany) using Vectashield^TM^ anti-fade mounting medium (Vector Laboratories, Newark, CA, USA). The samples were examined at the Multiuser Laboratory of Multiphoton Microscopy (LMMM) at FMRP-USP by the multiphoton microscopy imaging system using the Zeiss LSM 7MP Multiphoton Microscope (Carl Zeiss), using a W Plan-Apochromatica ×10 and ×40/1.0 objective (Carl Zeiss) and Ti: Saphire Chameleon Vision II laser systems (Coherent Inc., Easton, PA, USA). For the formation of 2D images, 30 images were acquired in a vertical plane along 100 µm of thickness. The quantification of ALP and BSP in different experimental groups was accessed in 2D images using the “Straight Lines” tool of the ImageJ software (public domain software, developed by Wayne Rasband-NIMH, NIH, Bethesda, MD, USA, http://rsbweb.nih.gov/ij, accessed on 15 January 2020). In each group, 50 cells were selected. The results were obtained from the fluorescence intensity of each cell.

### 2.6. Statistical Analysis

The data obtained were analyzed using GraphPad Prism software (5.0 version, GraphPad Software Inc.). Normality analyses of distribution data were carried out by the Shapiro-Wilk test (*p* < 0.05). To evaluate differences between two groups, Student’s t parametric or Mann–Whitney non-parametric tests were performed. The significance level was *p* < 0.05 for all tests. The results were expressed textually and in graphs with mean or median and standard error (mean ± standard error) or interquartile (median ± interquartile) shift.

## 3. Results

Figure 1 shows the grafting (A) and the removal after 16 h (B) of the SCA in 5 mm bone defects in the parietal bone of rats. Note the differential aspects of blood coloration at the time of grafting (intense red color) and immediately prior to removal (dark red color). These characteristics were observed in all animals.

### 3.1. Higher Prevalence of Platelets in PhC

Flow cytometry allowed the characterization of PhC cells using cluster of differentiation (CD) surface markers for mesenchymal stem cells (MSC—CD90); lymphocytes (CD45); hematopoietic progenitor cells (CD34); leukocytes (CD44); platelets (CD42); and for Von Willebrand factor, endothelial, myeloid, and osteoclastic cells (CD61) (Figure 2A). The PhC formed on SCA exhibited a prevalence of CD42 (8.35%, 8.35 ± 2.6), followed by CD44 (7%, 6.7 ± 1.56), CD45 (5.45%, 5.45 ± 6.44), CD90 (3.95%, 3.95 ± 1.2), CD61 (3.8%, 3.2 ± 1.0), and CD34 (2.15%, 2.15 ± 1.1) detection. These results showed that the PhC formed in SCA inserted in critical defects created in rat calvaria presented different cell types, including platelets, immunoinflammatory cells, and progenitors cells (Figure 2B).

### 3.2. Cells Viability Was More Evident When Phc Was Incorporated into the Sca

The MTT assay (Figure 2C) showed a higher prevalence of viable cells from cultured OSB over SCA with PhC (*p* = 0.0169, 1.153 ± 0.2241) compared to those without PhC (0.3718 ± 0.08076). This shows the fundamental role of PhC on the cell viability of OSB on SCA.

### 3.3. PhC Showed a Rich Fibrin Mesh

Scan electromicrographs acquired in SEM (Figure 2(D.1)–(D.3)) revealed that PhC elements persisted during the 3 days in cell culture. In the groups with OSB only, the cells adhered to the SCA filaments, and we were able to clearly see their pores made by the 3D printer (Figure 2(D.1)). In SCA with PhC, there were interactions between osteoblastic cells (Figure 2(D.2), arrow) and ex vivo blood clot elements (Figure 2(D.2), arrow head), leading to visible extracellular matrix deposition by OSB on PhC (Figure 2(D.2), asterisk). This deposited matrix was not observed in the SCA + PhC group, showing the PhC structure without OSB, with emphasis on white and red blood cells (Figure 2(D.3), arrow), inserted in a rich and dense fibrin meshwork (Figure 2(D.3), arrowhead), sealing all the SCA pores.

### 3.4. Ex Vivo Models May Delay the Mineralization Process

The mineralized matrix deposit formation was evaluated at 7 and 10 days of experiment. At 7 days (Figure 3A), there were no statistically significant differences between SCA + OSB (*p* = 0.5722; 0.7007 ± 0.05574) and SCA + PhC + OSB (0.6397 ± 0.08218). At 10 days (Figure 3B), there was more formation of mineralized matrix in OSB cultured on SCA without PhC (*p* = 0.0199, 0.8233 ± 0.04617) when compared to the PhC group (0.5287 ± 0.06345). This shows that the presence of PhC reduces mineralized matrix formation in an ex vivo model.

### 3.5. ALP and BSP Immunodetection Was Different in Each Group

The multiphoton microscopy provided 2D images of OSB seeded on 3D SCA without (Figure 4A–E) and with PhC (Figure 4G–K) at 10 days. ALP expression (Figure 4F) was more intense in SCA + OSB (*p* = 0.0021; 118.3 ± 30.61) than SCA + PhC + OSB group (79.05 ± 33.76). In both groups, ALP labeling (Figure 4C,I) presented a granular appearance in cytoplasm, allowing prompt cell morphology identification and localization of OSB cells on SCA. In contrast, BSP immunostaining (Figure 4L) was more intense in OSB seeded on PhC formed on SCA (*p* = 0.0256; 17.18 ± 11.31) when compared to the SCA + OSB group (4.187 ± 13.21). The BSP labeling (Figure 4D,J) was similar in both groups and showed a dispersed and irregular distribution on SCA.

## 4. Discussion

This study compared the features of osteoblastic cells on SCA in different environments—ex vivo and in vitro models. For the ex vivo assay development, a 3D chitosan-based/β-TCP SCA was inserted in a critical calvarial defect for 16 h for PhC formation. The flow cytometry (FC) findings showed platelet predominance in PhC, in addition to other white blood cells, such as leukocytes, lymphocytes, and progenitor cells. The SEM results corroborated the FC findings, indicating a extensive fibrin mesh formation interacting with different blood cell types, which persisted in the cell culture medium. In addition, the PhC increased cell viability, reduced mineralized matrix deposition and ALP expression in OSB + SCA group. These findings showed that the PhC addition on SCA in ex vivo assays clearly alters OSB behavior in a 3D microenvironment. Furthermore, it demonstrated the potential use of blood clots created in an experimental animal model for cell culture applications.

The blood clot displays a crucial role in tissue repair, coordinating numerous cells involved in the initial phase of inflammation and hematoma formation within the first 24 h of tissue injury [33]. Studies involving blood in ex vivo assays were already documented in the literature. In these studies, the individual’s blood was collected and applied to the surface material. To prevent blood coagulation in the collection tubes, anticoagulant substances such as heparin, EDTA, or citrate were added [4,34]. These agents directly impact the behavior of immune response cells, altering the performance of a key player in tissue repair, potentially biasing the results [35].

Several experimental models were tested to evaluate bone regeneration using SCA [36]. The critical-size defects in rat calvariua are among the most widely used models, as the technique is simple and cost-effective and produces standardized bone defects, facilitating laboratory analysis [37]. Additionally, the calvarium is supported by the dura mater and dermis during suturing, facilitating the biomaterial stabilization inserted in the surgical site [37].

Another key factor from a regenerative perspective is the embryological origin of the calvaria, as both neural crest and mesoderm cells simultaneously contribute to craniofacial intramembranous ossification [38,39]. Cranial neural crest cells possess a higher osteogenic potential compared to mesodermal cells in long bones [40,41]. Therefore, the use of this calvaria model likely facilitates cell recruitment from different embryological origins, making it a valuable model for studying osteoconductive biomaterials and conducting ex vivo assays.

In PhC formed on SCA, platelets were prevalent, with different white blood cells also present. Endothelial damage will lead to the exposure of collagen fibers, von Willebrand factor, and fibronectin, enabling the aggregation of main blood components, such as platelets and fibrin [7,42]. Platelets are essential in hemostasis and immunology [43]. During the formation of the hemostatic plug, granules enriched with chemokines are released, facilitating the recruitment of leukocytes [44]. This will lead to stable blood clot formation, enriched with progenitor and inflammatory cells embedded in a dense fibrin mesh [45], as observed in SEM findings. The components of the blood clot formed possibly contain numerous viable cells and may have contributed to OSB proliferation, as evidenced by the MTT results. Throughout the osteogenesis process, the hematoma and acute inflammation are gradually replaced by an ECM [45], as observed in the mineralization assays conducted at 7 and 10 days. At 10 days, there was more formation of mineralized matrix in the group without PhC. Due to its mechanical characteristics and composition being similar to ECM, chitosan was previously used for 3D SCA fabrication [46]. The association of this hydrogel with β-TCP potentiated osteogenesis in mesenchymal stem cells [47,48]. The addition of PhC in ex vivo assays must have delayed the osteoblastic differentiation generated by the biomaterial itself. The cells present must have been competitive in the microenvironment created in SCA, making it difficult to achieve the full differentiation state. This evidence was confirmed when ALP expression was quantified, as it was less expressed in the SCA + PhC + OSB group.

ALP is a metalloenzyme directly associated with the initiation of ECM mineralization mechanism and osteoblastic differentiation [49]. It is responsible for the deposition of hydroxyapatite in the collagen fibers of the bone matrix [50]. The ALP expression and the mineralized matrix formation were likely delayed by the presence of PhC. These findings demonstrated the feasibility of using PhC in ex vivo assays. Moreover, incorporating PhC into the experiments altered OSB behavior compared to in vitro assays in a 3D microenvironment. This “closer to nature” setting can significantly shift the outcomes of a study.

## 5. Conclusions

The use of PhC in ex vivo assays showed a behavioral difference in osteoblasts when compared to the in vitro study conducted on 3D SCA. The presence of blood clots in this study may have facilitated the interaction between white blood cells—typically present during inflammatory/repair response—and osteoblasts. This synergistic interaction in ex vivo assay suggests a behavior closer to the natural microenvironment, leading to differences response when compared to in vitro assays.

The differences observed between in vitro and ex vivo assays highlight potential biases associated with in vitro assays. The absence of a PhC in in vitro experiments creates a microenvironment that differs from in vivo conditions, where interactions between various cell types and the tissue matrix are crucial for new bone tissue formation. By incorporating a blood clot into ex vivo assays, the results more closely resemble the processes occurring in vivo.

Therefore, when analyses are conducted using ex vivo assays, the results more accurately reflect in vivo conditions, enabling more reliable and precise testing of biomaterials. This approach can streamline the process, accelerate the implementation of these biomaterials in clinical studies, and reduce the need for in vivo preclinical studies with animals.

## Figures and Tables

**Figure 1 biomimetics-09-00782-f001:**
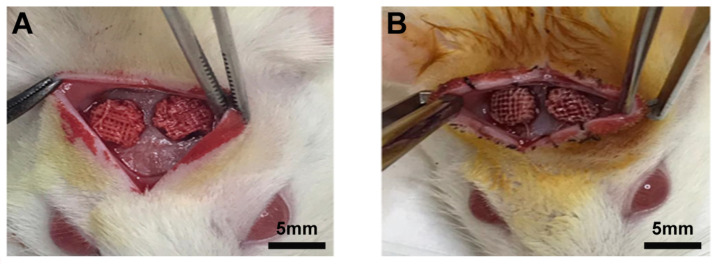
Images showing physiological blood clot (PhC) formation on scaffolds in a critical-size defect in rat calvaria. (**A**) The scaffold inserted immediately after the defect creation. (**B**) The scaffold with PhC formed after 16 h.

**Figure 2 biomimetics-09-00782-f002:**
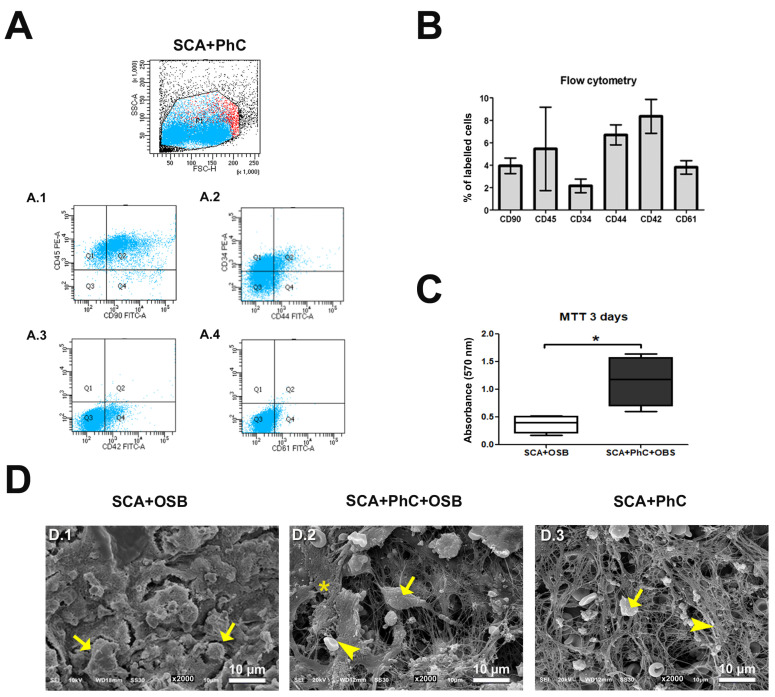
(**A**) Flow cytometry cell population distribution in physiological blood clot (PhC) formed on SCA, varying in size and complexity of cells for CD90/CD45 (**A.1**), CD44/CD34 (**A.2**), CD42 (**A.3**), and CD61 (**A.4**) in the histograms; (**B**) Proportion of labelled cells for CD90, CD45, CD34, CD44, CD42, and CD61 on PhC cell surface; (**C**) Cell viability results by MTT assay, expressed as absorbance at 570 nm; (**D**) Electron micrographs showing SCA + OSB (**D.1**), SCA + PhC + OSB (**D.2**), and SCA + PhC (**D.3**) at 2000× magnification. In the SCA + OSB group (**D.1**) the OSB cells (arrow) can be seen on the SCA surface. In the SCA + PhC + OSB group (**D.2**), there were interactions between OSB cells (arrow) and PhC elements (arrow head), leading to visible extracellular matrix deposition by OSB on PhC (asterisk). The SCA + PhC group (**D.3**) showing the PhC structure without OSB, with emphasis on white and red blood cells (arrow), inserted in a rich and dense fibrin meshwork (arrowhead), sealing all the SCA pores. Scale: 10 µm. Significance value: * *p* < 0.05.

**Figure 3 biomimetics-09-00782-f003:**
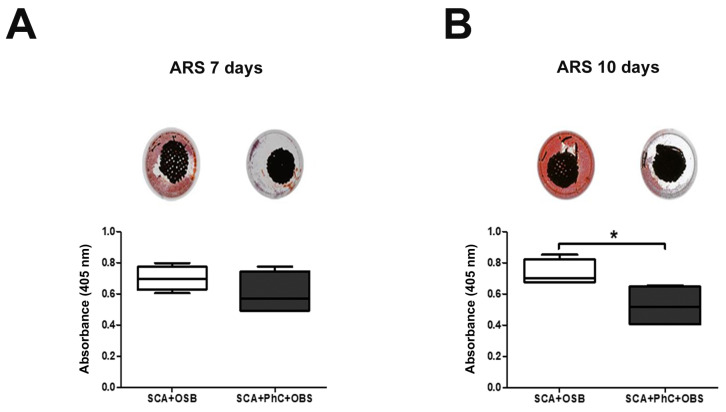
Mineralization assay at 7 days (**A**) and 10 days (**B**). The data on Alizarin red S (ARS) stain were expressed as absorbance at 405 nm. Significance value: * *p* <0.05.

**Figure 4 biomimetics-09-00782-f004:**
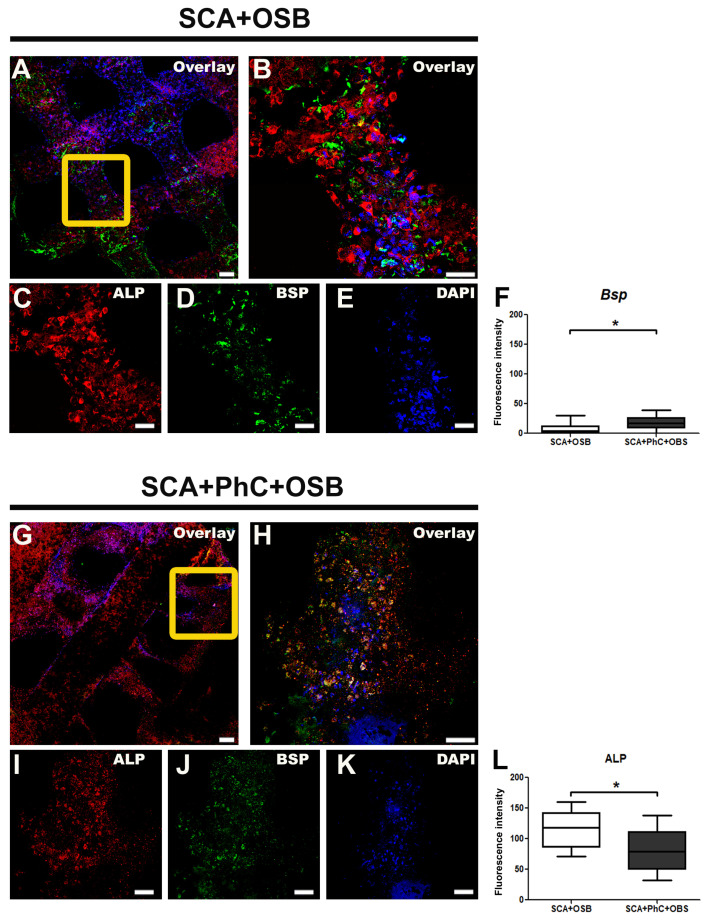
Immunofluorescence images obtained at Multiphoton Microscopy showing osteoblastic cells (OSBs) seeded on a scaffold (SCA) without (**A**–**E**) and with physiological blood clot (PhC) (**G**–**K**). The yellow squares in (**A**,**G**) represent the merged fluorescences in (**B**,**H**), respectively. Alkaline phosphatase (ALP) was labeled with Alexa Fluor 594 (**C**,**I**), bone sialoprotein (BSP) with Alexa Fluor 488 (**D**,**J**) and cell nuclei with DAPI (**E**,**K**). ALP (**F**) and BSP (**L**) immunoexpression between the SCA + OSB and SCA + PhC + OSB groups showed different results. Fluorescence intensity data (**F**,**L**) are expressed as arbitrary units of pixel intensity per area, as determined by ImageJ software. Scale bars: 20 µm in A and G, and 50 µm in (**B**–**E**,**H**–**K**). Significance value: * *p* < 0.05.

## Data Availability

The data presented in this study are available upon request from the corresponding author under plausible justification.

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
