# Peer review of "The Influence of Physiological Blood Clot on Osteoblastic Cell Response to a Chitosan-Based 3D Scaffold—A Pilot Investigation"

_biomimetics, 2024, doi:10.3390/biomimetics9120782_

Round 1
Reviewer 1 Report
Comments and Suggestions for Authors
General Comments:
This manuscript is an interesting approach to evaluate the impact of physiological coagulation (PhC) on osteogenesis using a 3D scaffold based on chitosan and beta-tricalcium phosphate. The authors compared the scaffolds loaded/seeded with osteoblasts (OSB) and scaffolds loaded with osteoblasts and PhC. Obtained results showed that PhC increased cell viability, reduced mineralized matrix deposition and alkaline phosphatase (ALP) expression in scaffolds with OSB. These findings highlight the potential of using PhC in ex vivo assays to design biomaterials with a behavior closer to the natural microenvironment for cell response. The manuscript is well written, with examples and illustration. This topic should be of interest for the community of medicine, tissue engineering, and biomaterials science so I recommend its publication.
Author Response
The authors thank Reviewer 1 for his/her overall positive perception of the manuscript and important comment.
Comments 1: "This manuscript is an interesting approach to evaluate the impact of physiological coagulation (PhC) on osteogenesis using a 3D scaffold based on chitosan and beta-tricalcium phosphate. The authors compared the scaffolds loaded/seeded with osteoblasts (OSB) and scaffolds loaded with osteoblasts and PhC. Obtained results showed that PhC increased cell viability, reduced mineralized matrix deposition and alkaline phosphatase (ALP) expression in scaffolds with OSB. These findings highlight the potential of using PhC in ex vivo assays to design biomaterials with a behavior closer to the natural microenvironment for cell response. The manuscript is well written, with examples and illustration. This topic should be of interest for the community of medicine, tissue engineering, and biomaterials science so I recommend its publication."
Answer: The authors appreciate the valuable insights regarding the paper. The study conducted will be of fundamental importance to the scientific community, contributing to the alignment of ex vivo results with the conditions observed in in vivo microenvironment. The presence of the blood clot will allow for the joint interaction of immune-inflammatory response cells and progenitor cells in laboratory assays.
Reviewer 2 Report
Comments and Suggestions for Authors
What was the reasoning behind selecting 16 hours as the optimal time for physiological blood clot (PhC) formation? Were other time points tested or considered?
Could more details be provided on the selection of the 200 µm pore size for the 3D scaffold? How might this pore size affect the interactions between osteoblasts and the scaffold?
What factors influenced the choice of the UMR-106 osteoblastic cell lineage for the experiments? Would similar results be expected with different cell types?
Could the delayed mineralization observed in the ex vivo model have implications for scaffold design in clinical settings? If so, how might this influence future studies?
How might the presence of platelets and immune cells in the PhC affect osteoblast behavior, particularly regarding viability and differentiation?
Was there any attempt to quantify the specific growth factors or cytokines released by the blood clot? How could these have influenced the experimental outcomes?
The study shows reduced mineralized matrix formation in the PhC group after 10 days. What mechanisms could be responsible for the delay in mineralization in the presence of PhC?
Was the structural integrity of the scaffold over time, particularly with PhC formation, investigated? If not, what potential degradation or weakening of the scaffold might be anticipated?
How do the differences observed between in vitro and ex vivo models inform the transition to in vivo studies or potential clinical applications?
Are there plans to conduct longer-term studies to determine whether mineralization in the ex vivo model eventually matches or surpasses that in the in vitro model at later time points?
Author Response
Comments 1: What was the reasoning behind selecting 16 hours as the optimal time for physiological blood clot (PhC) formation? Were other time points tested or considered?
Answer: We conducted testing for 16 hours, following the adapted protocol established by Monroe et al. (2012) and Zuardi et al. (2023). This time frame was also chosen to allow for physiological coagulum formation, facilitating the migration and interaction of various cell types, particularly mesenchymal cells. This interaction triggers a cascade of critical events that contribute to the formation of a matrix. Consequently, the matrix formed within the scaffold can, when used for ex vivo studies, mimic a microenvironment similar to in vivo conditions.
References:
Monroe, D.M.; Hoffman, M. The clotting system–a major player in wound healing. Haemophilia 2012, 18, 11–16.
Zuardi, L.R.; Silva, C.L.A.; Rego, E.M.; Carneiro, G.V.; Spriano, S.; Nanci, A.; de Oliveira, P.T. Influence of a Physiologically Formed Blood Clot on Pre-Osteoblastic Cells Grown on a BMP-7-Coated Nanoporous Titanium Surface. Biomimetics 2023, 8, 123.
Comments 2: Could more details be provided on the selection of the 200 µm pore size for the 3D scaffold? How might this pore size affect the interactions between osteoblasts and the scaffold?
Answer: Although osteoblasts range in size from 10 to 50 μm (Sugawara et al., 2005), a 200 μm porosity was selected for the 3D scaffold because this pore size is optimal for tissue mineralization during bone regeneration. Moreover, this porosity facilitates macrophage migration, which contributes to pathogen elimination and plays a direct role in osteoimmunology and blood vessel formation (Iviglia et al., 2019).
References:
Sugawara, Y.; Kamioka, H.; Honjo, T.; Tezuka, K.; Takano-Yamamoto, T. Three-dimensional reconstruction of chick calvarial osteocytes and their cell processes using confocal microscopy. Bone 2005, 36, 877-83.
Iviglia, G.; Kargozar, S.; Baino, F. Biomaterials, current strategies, and novel nano-technological approaches for periodontal regeneration. J. Funct. Biomater. 2019, 10.
Comments 3: What factors influenced the choice of the UMR-106 osteoblastic cell lineage for the experiments? Would similar results be expected with different cell types?
Answer: The UMR-106 cell line is widely used in in vitro models to study normal bone biology (Imai et al., 1998; Koide et al., 2017; Takahashi et al., 2017; Pierrefite-Carle et al., 2017; Zhuardi et al., 2023). According to Rodan and Noda (1991) and Imai et al. (1998), UMR-106 cells are considered pre-osteoblasts, serving as immediate precursors to fully differentiated osteoblasts due to their osteoblastic characteristics and morphological features, which closely resemble those of osteoblastic lineage cells observed in in vivo.
In our recent study, we examined the effects of aptamers on blood clot formation and the UMR-106 cells behavior (da Costa et al., 2023). The findings demonstrated that the aptamer significantly increased alkaline phosphatase (ALP) and bone sialoprotein (BSP) immunolabeling in osteoblastic cells. Furthermore, the morphological characteristics of ALP and BSP labeling were consistent with those typically seen in osteoblastic cells.
Thus, UMR-106 cells provide a reliable model for studying normal bone biology in vitro. This cell line, derived from a rat clonal osteosarcoma, exhibits osteoblast-like phenotypic and behavioral properties, making it a valuable tool for research in bone biology.
References:
Imai, S.; Kaksonen, M.; Raulo, E.; Kinnunen, T.; Fages, C.; Meng, X.; Lakso, M.; Rauvala, H. Osteoblast recruitment and boné formation enhanced by cell matrix–associated heparin-binding growth-associated molecule (HB-GAM). The Journal of cell biology 1998, 143, 1113–1128.
Rodan, G.; Noda, M. Gene expression in osteoblastic cells. Critical reviews in eukaryotic gene expression 1991, 1, 85–98.
Koide, M.; Kobayashi, Y.; Yamashita, T.; Uehara, S.; Nakamura, M.; Hiraoka, B.Y.; Ozaki, Y.; Iimura, T.; Yasuda, H.; Takahashi, N.; et al. Bone formation is coupled to resorption via suppression of sclerostin expression by osteoclasts. Journal of Bone and Mineral Research 2017, 32, 2074–2086.
Pierrefite-Carle, V.; Santucci-Darmanin, S.; Breuil, V.; Gritsaenko, T.; Vidaud, C.; Creff, G.; Solari, P.L.; Pagnotta, S.; Al-Sahlanee, R.; Auwer, C.D.; et al. Effect of natural uranium on the UMR-106 osteoblastic cell line: impairment of the autophagic process as an underlying mechanism of uranium toxicity. Archives of toxicology 2017, 91, 1903–1914.
Zuardi, L.R.; Silva, C.L.A.; Rego, E.M.; Carneiro, G.V.; Spriano, S.; Nanci, A.; de Oliveira, P.T. Influence of a Physiologically Formed Blood Clot on Pre-Osteoblastic Cells Grown on a BMP-7-Coated Nanoporous Titanium Surface. Biomimetics 2023, 8, 123.
Comments 4: Could the delayed mineralization observed in the ex vivo model have implications for scaffold design in clinical settings? If so, how might this influence future studies?
Answer: Delayed mineralization will not influence the future design of the 3D scaffold used in the experiments. In this research, delayed mineralization was result of the blood clot presence in scaffold, possibly because there are other cell types competing for the same cellular microenvironment.
Comments 5: How might the presence of platelets and immune cells in the PhC affect osteoblast behavior, particularly regarding viability and differentiation?
Answer: Platelets play a crucial role in the initial stages of tissue repair, as they are responsible for releasing various growth factors that promote the recruitment of regenerative cells and initiate the immune response simultaneously (Locatelli et al., 2021). In an in vivo environment, regenerative cells typically face no significant challenge, given the abundance of nutrient sources near the repair site. However, in an ex vivo environment, where nutrient supply is controlled via cell culture, this can pose a challenge for osteoblast cells.
As observed by scanning electron microscopy in Figure 2, the physiological coagulum formed in 3D scaffold contained multiple cell types embedded in a rich fibrin network. These cell types likely competed within the same microenvironment, potentially inducing cellular stress in the osteoblasts. Although oxidative stress was not specifically analyzed in the studied groups, this observation opens new avenues for future investigations.
References:
Locatelli, L.; Colciago, A.; Castiglioni, S.; Maier, J.A. Platelets in Wound Healing: What Happens in Space? Front. Bioeng. Biotechnol. 2021, 1-11.
Comments 6: Was there any attempt to quantify the specific growth factors or cytokines released by the blood clot? How could these have influenced the experimental outcomes?
Answer: Dear reviewer, thank you for your valuable feedback. As this is a pilot study, our primary focus was to examine the morphological characteristics of blood clot formed and to evaluate the specific effects of the substrate generated within the scaffold on osteoblast behavior. The quantification of growth factors and cytokines will be considered in future studies conducted by our research group.
Comments 7: The study shows reduced mineralized matrix formation in the PhC group after 10 days. What mechanisms could be responsible for the delay in mineralization in the presence of PhC?
Answer: In this context, the beta-tricalcium phosphate (β-TCP) present in the scaffold may have been a key factor contributing to the observed differences. In the SCA+OSB group, osteoblasts directly interacted with scaffold filaments enriched with β-TCP, which has been shown to enhance the osteogenic potential of osteoblastic cells (Sun et al., 1997). This direct interaction likely promoted the formation of a mineralized matrix. Furthermore, as highlighted in the discussion, the presence of other cell types within the physiological coagulum may have hindered osteoblastic differentiation.
References:
Sun, J.S.; Tsuang, Y.H.; Liao, C.J.; Liu, H.C.; Hang, Y.S.; Lin, F.H. The effects of calcium phosphate particles on the growth of osteoblasts. J Biomed Mater Res. 1997, 37, 324-34.
Comments 8: Was the structural integrity of the scaffold over time, particularly with PhC formation, investigated? If not, what potential degradation or weakening of the scaffold might be anticipated?
Answer: Your comment is greatly appreciated. Unfortunately, no specific analysis was conducted in this regard. However, throughout the experiments, the scaffold maintained its structural integrity, preserving both its porosity and morphology, which ensured the successful completion of all analyses.
Comments 9: How do the differences observed between in vitro and ex vivo models inform the transition to in vivo studies or potential clinical applications?
Answer: The differences observed between in in vitro and ex vivo assays highlight potential biases associated with in vitro assays. The blood clot absence in in vitro experiments creates a microenvironment that differs from the one seen in in vivo, where interactions between various cell types in the tissue matrix are crucial for new bone tissue formation. By introducing a blood clot into the ex vivo assays, the results more closely resemble the processes that occur in in vivo conditions.
Comments 10: Are there plans to conduct longer-term studies to determine whether mineralization in the ex vivo model eventually matches or surpasses that in the in vitro model at later time points?
Answer: Yes, there are plans for future analyses. The research group intends to investigate mineralization over extended periods, such as 14 and 21 days.
Reviewer 3 Report
Comments and Suggestions for Authors
The authors present a research article about the response of osteoblast to chitosan-based 3D-printed scaffolds. The idea is interesting. Unfortunately, the study design is lacking. This study focuses too much on the in vitro part and completely disregards the characterization of the scaffolds. This manuscript should be rejected, since it is lacking and questionable at all. The whole manuscript seems clapped together. I got the impression that the study cannot be produced, since a lot of information are missing in the Materials and Methods section.
1. Introduction: The introduction is lacking of the discussion of other biodegradable polymers that could be also used. Moreover, what about biostable polymers, that are non-biodegradable? How can they be used for the same application?
2. Line 26-27: I disagree. What about other composite scaffolds that has been investigated already? The guidance for different cell types has also been investigated by the appliance of different infill pattern in 3D printed structures. What about different surface modifications for these kind of structures, which also has been investigated? This is all missing and should be discussed.
3. The use of abbreviations their definitions is lacking, even in the figure captions.
4. State the used chemicals correctly. (purity, manufacturer, city of manufacturing, country of manufacturing).
5. State the used devices correctly: (type, manufacturer, city of manufacturing, country of manufacturing). It is highly doubtful that Zeiss is a Canadian company.
6. State the used software correctly: (version, developer, city of developing, country of developing).
7. For the 3D printing. What printer and which printing parameter has been used? What was the printing mechanism applied? How where the printing parameters been chosen? Printing temperature, printing speed, noozle size, printing-table temperature?
8. 3D printed scaffolds: What is the pore size, the infill pattern, the printed filament diameter? What are the dimensions of the scaffolds?
9. For the used chitosan: What is the molecular weight and the degree of acetylation? How was the chitosan dissolved?
10. Where filaments used for the 3D printing process? How was the calcium phosphate incorporated in the chitosan matrix?
11. Line 56: (Carl Zeiss, ALE)? What is ALE?
12. Line 38-39: According to the ARRIVE guidelines, the conditions for the hosting of the animals should be provided and why this animal type has been selected. Moreover, the selection of the animals and the diving in separate groups must be provided. The treatment and surgery of the animals should also be described in accordance to the guidelines. Why the ARRIVE 2.0 guidelines where not applied?
13. Figure 2: What is the meaning of the arrows and stars? This information is not provided in the figure caption. Significance value: * p <0.05, ** p <0.01, *** p <0.001 applying to what? This is also not clear.
14. According to Figure 2D1: What is the roughness of the scaffold surface?
15. What is the chemical stability of the used chitosan/CaP composite after 3D printing? What are the crystallinity changes in the composite before and after 3D printing?
16. What about histograms from the interface between bone and scaffolds?
17. Conclusion: An outlook is missing. What are the important results and how can they be used in the future?
Author Response
Comments 1. Introduction: The introduction is lacking of the discussion of other biodegradable polymers that could be also used. Moreover, what about biostable polymers, that are non-biodegradable? How can they be used for the same application?
Answer: Dear Reviewer, we appreciate your valuable feedback. In response to your suggestion, we have added three paragraphs to the introduction discussing biodegradable and non-biodegradable polymers. All changes have been highlighted in yellow in the revised text.
“Several materials have been developed for the preparation of these 3D scaffolds (Abbasi et al., 2020). Among them are biodegradable polymers, such as hydrogel (Cao et al., 2021).
Hydrogels are highly hydrophilic compounds with excellent biological properties. They can be composed of natural polymers, such as those derived from extracellular matrix (ECM) components (e.g., collagen, gelatin, fibrin, hyaluronic acid) or seaweed sources (e.g., alginate, agarose, chitosan). Alternatively, they can be made from chemical or synthetic polymers, including those derived from ethylene oxide, vinyl alcohol, acrylic acid, propylene fumarate-co-ethylene glycol, and polypeptides (Cao et al., 2021).
In this context, natural biodegradable polymers are the preferred choice over non-biodegradable ones due to their reduced toxicity, rapid biodegradation, osteogenic potential, and high biocompatibility. Conversely, non-biodegradable polymers, while associated with some toxicity and inflammatory responses, offer a slower degradation rate and superior physical-mechanical properties (Wei et al., 2020).
Among natural hydrogels, chitosan has been extensively studied for its potential in developing 3D scaffolds for bone regeneration (Parisi et al., 2017). Due to its mechanical properties and composition resembling the extracellular matrix (ECM), chitosan has proven to be an excellent biomaterial for creating membranes used in guided tissue regeneration in periodontics (Qasim et al., 2017). Moreover, it exhibits effective antimicrobial activity against various oral pathogens (Costa et al., 2012).”
References:
Cao, H.; Duan, L.; Zhang, Y.; Cao, J.; Zhang, K. Current hydrogel advances in physicochemical and biological response-driven biomedical application diversity. Signal Transduction and Targeted Therapy 2021, 6, 1-31.
Wei, S.; Ma, J.X.; Xu, L.; Gu, X.S.; Ma, X.L. Biodegradable materials for bone defect repair. Military Medical Research 2020, 7, 1-25.
Abbasi, N.; Hamlet, S.; Love, R.M.; Nguyen, N.T. Porous scaffolds for bone regeneration. Journal of Science: Advanced Materials and Devices 2020, 5, 1-9.
Parisi, L.; Galli, C.; Bianchera, A.; Lagonegro, P.; Elviri, L.; Smerieri, A.; Lumetti, S.; Manfredi, E.; Bettini, R.; Macaluso, G. Anti-fibronectin aptamers improve the colonization of chitosan films modified with D-(+) Raffinose by murine osteoblastic cells. Journal of Materials Science: Materials in Medicine 2017, 28, 1–12.
Qasim, S.B.; Najeeb, S.; Delaine-Smith, R.M.; Rawlinson, A.; Rehman, I.U. Potential of electrospun chitosan fibers as a surface layer in functionally graded GTR membrane for periodontal regeneration. Dental Materials 2017, 33, 71–83.
Costa, E.; Silva, S.; Pina, C.; Tavaria, F.; Pintado, M. Evaluation and insights into chitosan antimicrobial activity against anaerobic oral pathogens. Anaerobe 2012, 18, 305–309.
Comments 2. Line 26-27: I disagree. What about other composite scaffolds that has been investigated already? The guidance for different cell types has also been investigated by the appliance of different infill pattern in 3D printed structures. What about different surface modifications for these kinds of structures, which also has been investigated? This is all missing and should be discussed.
Answer: Dear Reviewer, we agree with your argument. All changes, as described in Comment 1, have been made and are highlighted in yellow in the Introduction section.
Comments 3. The use of abbreviations their definitions is lacking, even in the figure captions.
Answer: We agree with the reviewer. All abbreviations have been added and are highlighted in yellow throughout the text and figure legends.
Comments 4. State the used chemicals correctly (purity, manufacturer, city of manufacturing, country of manufacturing).
Answer: We agree with the reviewer. Data related to chemicals have been modified and are highlighted in yellow in the Materials and Methods section.
Comments 5. State the used devices correctly: (type, manufacturer, city of manufacturing, country of manufacturing). It is highly doubtful that Zeiss is a Canadian company.
Answer: We agree with the reviewer. The suggested modifications have been made and are highlighted in yellow in the Materials and Methods section.
Comments 6. State the used software correctly: (version, developer, city of developing, country of developing).
Answer: We agree with the reviewer.
The suggested modifications have been implemented and are highlighted in yellow in the Materials and Methods section.
Comments 7. For the 3D printing. What printer and which printing parameter has been used? What was the printing mechanism applied? How where the printing parameters been chosen? Printing temperature, printing speed, nozzle size, printing-table temperature?
Answer: We thank the reviewer for the question. We added a brief description of the 3D printing process used and a reference to improve the clarity of the experimental process:
“The ink solution was loaded into a 10 ml syringe that was accommodated on a 3D printer built in-house (Elviri et al., 2017). Briefly, the instrument exploits a three cartesian axes system that allows the movement on the horizontal plane of the printing plate and the vertical translation of the nozzle. The chitosan-based solution is printed by a pump-acting extrusion process throughout a syringe mounting a 26 G needle. A layer-by-layer deposition mode is carried out on the printing plate at a velocity of 3 mm/s and instantaneously frozen (-10°C) to keep the scaffold structure. Scaffolds (1.6 cm × 1.6 cm in size) were composed by the alternation of orthogonal layers of parallel filaments being 200 μm distant from each other.
After freezing of the 3D printed scaffold, the structure was fixed by exposition to ammonia vapor in a saturated chamber.”
The section described above has been added to the Materials and Methods, under the 3D Chitosan/β-TCP Scaffolds section.
References:
Elviri, L.; Foresti, R.; Bergonzi, C.; Zimetti, F.; Marchi, C.; Bianchera, A.; Bernini, F.; Silvestri, M.; Bettini, R. Highly defined 3D printed chitosan scaffolds featuring improved cell growth. Biomedical Materials 2017, 12, 045009.
Comments 8. 3D printed scaffolds: What is the pore size, the infill pattern, the printed filament diameter? What are the dimensions of the scaffolds?
Answer: As described in Comment 7, the scaffolds (1.6 cm × 1.6 cm in size) were composed by the alternation of orthogonal layers of parallel filaments being 200 μm distant from each other (Elviri et al., 2017).
References:
lviri, L.; Foresti, R.; Bergonzi, C.; Zimetti, F.; Marchi, C.; Bianchera, A.; Bernini, F.; Silvestri, M.; Bettini, R. Highly defined 3D printed chitosan scaffolds featuring improved cell growth. Biomedical Materials 2017, 12, 045009.
Comments 9. For the used chitosan: What is the molecular weight and the degree of acetylation? How was the chitosan dissolved?
Answer: We agree with the reviewer that the information is important for the readers and the clarity of the text. This suggestion has been incorporated into the text, in the Materials and Methods, in 3D Chitosan/β-TCP Scaffolds section.x
“In this work Chitosan ChitoClear™ TM4030, having a degree of deacetylation of 75% and a molecular weight of 50 kDa (Primex Ehf, Iceland) was used. Acetic acid was obtained by Sigma-Aldrich, USA. All solutions were prepared using ultrapure water obtained with a Purelab Flex 1 system by ELGA Veolia.
Chitosan was dissolved in ultrapure water at 6% w/v concentration with the help of 2% v/v of acetic acid under stirring. After complete dissolution of chitosan, the calcium phosphate was added and mixed until complete homogenization before printing. “
Comments 10. Where filaments used for the 3D printing process? How was the calcium phosphate incorporated in the chitosan matrix?
Answer: As described in comment 7, the chitosan-based solution is printed by a pump-acting extrusion process throughout a syringe mounting a 26 G needle. A layer-by-layer deposition mode is carried out on the printing plate at a velocity of 3 mm/s and instantaneously frozen (-10°C) to keep the scaffold structure. After freezing of the 3D printed scaffold, the structure was fixed by exposition to ammonia vapor in a saturated chamber.
After complete chitosan dissolution, the calcium phosphate was added and mixed until complete homogenization before printing.
Comments 11. Line 56: (Carl Zeiss, ALE)? What is ALE?
Answer: We appreciate the observation. There was a typing error, and the abbreviation was changed in the text.
Comments 12. Line 38-39: According to the ARRIVE guidelines, the conditions for the hosting of the animals should be provided and why this animal type has been selected. Moreover, the selection of the animals and the diving in separate groups must be provided. The treatment and surgery of the animals should also be described in accordance to the guidelines. Why the ARRIVE 2.0 guidelines where not applied?
Answer: We agree with the reviewer that this information is important for readers and contributes to the clarity of the text. The data related to the hosting conditions and the rationale for using Hannover rats have been added to the Ethics Aspects section and are highlighted in yellow. Regarding the surgical procedure, all steps are thoroughly detailed in the "Physiological" Blood Clot Formation section.
“In this stage, 42 male Hannover rats, aged 12 weeks, purchased from the Central Animal Facility of the City Hall of the USP Campus in Ribeirão Preto were used. These animals were selected because they are anatomically, physiologically, and genetically similar to humans (Clements et al., 2022). The sample size calculation was performed using the Graphpad Prism 5.0 program (GraphPad Software®). The ideal sample size used to ensure 80% power in the statistical analysis of the data obtained in this study was calculated considering the differences in means and standard deviation, with the α value adjusted to 0.05, using the study by Zuardi et al. 2023 as a reference. The animals were randomly housed in plastic vivariums, with 4 rats per box. During the housing period, the animals were fed with balanced food and water ad libitum, remaining at a temperature of 25ºC, in a 12 h dark/light cycle.”
References:
Zuardi, L.R.; Silva, C.L.A.; Rego, E.M.; Carneiro, G.V.; Spriano, S.; Nanci, A.; de Oliveira, P.T. Influence of a Physiologically Formed Blood Clot on Pre-Osteoblastic Cells Grown on a BMP-7-Coated Nanoporous Titanium Surface. Biomimetics 2023, 8, 123.
Clements, P.J.; Bolon, B.; McInnes, E.; Mukaratirwa, S.; Scudamore, C. Chapter 17 - Animal Models in Toxicologic Research: Rodents. In Haschek and Rousseaux’s Handbook of Toxicologic Pathology (Fourth Edition), Fourth Edition ed.; Haschek, W.M.; Rousseaux, C.G.; Wallig, M.A.; Bolon, B., Eds.; Academic Press, 2022; pp. 653–6
Comments 13. Figure 2: What is the meaning of the arrows and stars? This information is not provided in the figure caption. Significance value: * p <0.05, ** p <0.01, *** p <0.001 applying to what? This is also not clear.
Answer: We agree with the reviewer. Some electron micrographs images were removed because they were similar to those published in a previous article by our research group. In addition, the Figure 2 caption was modified in relation to the electron micrographs of the groups analyzed, and the arrows, arrowheads and asterisks description are cited according to their respective images. *p < 0.05 means that the asterisk in Figure 2.C graph shows the p value significance in the comparison performed in MTT assay, showing that there was a difference in the statistical comparison. ** p < 0.01 and *** p < 0.001 were also removed from the Figure 2 caption.

Figure 2. (A) - Flow cytometry cell population distribution in PhC formed on SCA, varying in size and complexity of cells for CD90/CD45 (A.1), CD44/CD34 (A.2), CD42 (A.3) and CD61 (A.4) in the histograms; (B) - Intensity of Fluorescence detected for CD90, CD45, CD34, CD44, CD42 and CD61 on PhC cells surface; (C) - Cell viability results by MTT assay; (D) Electromicrographs showing SCA+OSB (D.1), SCA+PhC+OSB (D.2) and SCA+PhC (D.3) at 2000x magnification. In SCA+OSB group (D.1), the OSB cells (arrow) can be seen on SCA surface. In SCA+PhC+OSB group (D.2) there were interactions between OSB cells (arrow) and PhC elements (arrow head), leading to visible extracellular matrix deposition by OSB on PhC formed (asterisk). The SCA+PhC group (D.3) showing the PhC structure without OSB, with emphasis on white and red blood cells (arrow), inserted in a rich and dense fibrin meshwork (arrowhead), sealing all the SCA pores. Scale: 10 μm. Significance value: * p <0.05.
Comments 14. According to Figure 2D1: What is the roughness of the scaffold surface?
Answer: The roughness observed on scaffold surface are irregularities present in filaments. In addition, the cellular adhesion of osteoblasts must have increased due to the cellular activities on scaffold filaments.
Comments 15. What is the chemical stability of the used chitosan/CaP composite after 3D printing? What are the crystallinity changes in the composite before and after 3D printing?
Answer: Unfortunately, we do not have results for this data at the moment, but it will be the focus of our future investigations.
Comments 16. What about histograms from the interface between bone and scaffolds?
Answer: Unfortunately, we do not have results for this data at the moment, but it will be the focus of our future investigations.
Comments 17. Conclusion: An outlook is missing. What are the important results and how can they be used in the future?
Answer: We agree with the reviewer. Revisions have been made in Conclusion section and are highlighted in yellow.
“The differences observed between in vitro and ex vivo assays highlight potential biases associated with in vitro assays. The absence of a PhC in in vitro experiments creates a microenvironment that differs from in vivo conditions, where interactions between various cell types and the tissue matrix are crucial for new bone tissue formation. By incorporating a blood clot into ex vivo assays, the results more closely resemble the processes occurring in vivo.
Therefore, when analyses are conducted using ex vivo assays, the results more accurately reflect in vivo conditions, enabling more reliable and precise testing of biomaterials. This approach can streamline the process, accelerate the implementation of these biomaterials in clinical studies, and reduce the need for in vivo preclinical studies with animals.”
Round 2
Reviewer 3 Report
Comments and Suggestions for Authors
The authors improved their manuscript, but there are still some points left. The following points should be addressed in the next revision round:
1. All chemicals used are still not stated correctly: (grade/purity, company, city of production, country of production).
2. All used devices are still not stated correctly: (type, company, city of production, country of production). For example: chapter 2.2 – What 3D printer?
3. (Carl Zeiss,Oberkochen, DEU), (Knittel Glass, Braunschweig, DEU) and (Leica Microsystems, Wetzlar, DEU) – Which country is in English “DEU”? Better to write the complete country names in English.
4. All software used are still not stated correctly: (version, developer, city of development, country of development).
5. It should be also explained why Hannover rats have been used and not Wistar rats, for example.
6. Why is Figure 1 in chapter 2 when it shows results? The results of implantation. Moreover, the images should have a scale bar that it can be understood what is the real size of implants.
7. “Then, the PhC was gently washed with PB (Gibco, Invitrogen®)...” – What is PB here?
8. Figure 2B, 4F and 4L: What is IF? What is the unit? Since there is a numbering at the y-axis, it should be related to the intensity at the wavelength of the fluorescence of emission or a real unit.
9. Figure 2C: nm for MTT assay? OD at the applied wavelength is here common. Or absorbance, what is it.
10. Figure 2D: The scale bars should be redrawn to a readable size. I can't really read it even when I zoom in.
11. Figure 4: The figure caption should state the meaning of the yellow box. “Scale: 20 µM and 50 µM” – For which images? The reviewer and reader should guess by themselves?
12. Conclusion: In vitro, in vivo and ex vitro should be written correctly.
13. In all figure captions and in every paragraph, all used abbreviations should be defined. Moreover, some are never defined. Or make an overview of all abbreviations used in this study. It is exhausting to look up the meanings of the abbreviations every time again and again. As a result, the manuscript quickly becomes difficult to read.
Author Response
Comments 1. The authors improved their manuscript, but there are still some points left. The following points should be addressed in the next revision round: All chemicals used are still not stated correctly: (grade/purity, company, city of production, country of production).
Answer: We agree the reviewer and changed the text accordingly.
Comments 2. All used devices are still not stated correctly: (type, company, city of production, country of production). For example: chapter 2.2 – What 3D printer?
Answer: The reviewer has a point. The text was then modified to state all the devices correctly.
Comments 3. (Carl Zeiss,Oberkochen, DEU), (Knittel Glass, Braunschweig, DEU) and (Leica Microsystems, Wetzlar, DEU) – Which country is in English “DEU”? Better to write the complete country names in English.
Answer: We agree with the reviewer. Complete country names are now in English.
Comments 4. All software used are still not stated correctly: (version, developer, city of development, country of development).
Answer: The text has been modified accordingly.
Comments 5. It should be also explained why Hannover rats have been used and not Wistar rats, for example.
Answer: We thank the reviewer about this recommendation. The correct name is Wistar Hannover and it is now stated in the text. In addition, the rationale for using this rat strain now appears in the text.
Comments 6. Why is Figure 1 in chapter 2 when it shows results? The results of implantation. Moreover, the images should have a scale bar that it can be understood what is the real size of implants.
Answer: We agree with the reviewer. Figure 1 is now in the Results section.
Comments 7. “Then, the PhC was gently washed with PB (Gibco, Invitrogen®)...” – What is PB here?
Answer: PB stands for Phosphate Buffer, which is now stated in the text.
Comments 8. Figure 2B, 4F and 4L: What is IF? What is the unit? Since there is a numbering at the y-axis, it should be related to the intensity at the wavelength of the fluorescence of emission or a real unit.
Answer: The reviewer has a point. Regarding Figure 2B, the y-axis represents proportion of labelled cells for CD90, CD45, CD34, CD44, CD42 and CD61 on PhC cells surface. For Figure 4F and 4L, at the y-axis we now indicate Fluorescence Intensity, which is expressed by arbitrary units of pixel intensity per area, as determined by ImageJ software. All changes are highlighted in yellow in the Figure 4 caption.
Comments 9. Figure 2C: nm for MTT assay? OD at the applied wavelength is here common. Or absorbance, what is it.
Answer: We agree with the reviewer. It is absorbance at 570 nm. This is now highlighted in yellow in the Figure 2 caption.
Comments 10. Figure 2D: The scale bars should be redrawn to a readable size. I can't really read it even when I zoom in.
Answer: The reviewer has a point. The Figure 2D was changed accordingly.
Comments 11. Figure 4: The figure caption should state the meaning of the yellow box. “Scale: 20 µM and 50 µM” – For which images? The reviewer and reader should guess by themselves?
Answer: The reviewer has a point. The figure legend for Figure 4 is now changed accordingly. The changes are highlighted in yellow in the Figure 4 caption.
Comments 12. Conclusion: In vitro, in vivo and ex vitro should be written correctly.
Answer: We agree with the reviewer. The changes are highlighted in yellow in the text.
Comments 13. In all figure captions and in every paragraph, all used abbreviations should be defined. Moreover, some are never defined. Or make an overview of all abbreviations used in this study. It is exhausting to look up the meanings of the abbreviations every time again and again. As a result, the manuscript quickly becomes difficult to read.
Answer: We reviewed all the abbreviations and defined then.